# Wetter summers can intensify departures from natural variability in a warming climate

Colin R. Mahony[1] & Alex J. Cannon [2]

Climate change can drive local climates outside the range of their historical year-to-year variability, straining the adaptive capacity of ecological and human communities. We demonstrate that dependencies between climate variables can produce larger and earlier departures from natural variability than is detectable in individual variables. Using the example of summer temperature (Tx) and precipitation (Pr), we show that this departure intensification effect occurs when the bivariate climate change trajectory is misaligned with the dominant mode of joint historical variability. Departure intensification is evident in all six CMIP5 models that we examined: 23% (9–34%) of the global land area of each model exhibits a pronounced increase in $2\sigma$ anomalies in the Tx-Pr regime relative to Tx or Pr alone. Observational data suggest that summer Tx-Pr correlations in distinct regions on all continents are sufficient to produce departure intensification. Precipitation can be an important driver of multivariate climate change signals relative to natural variability, despite typically having a much weaker univariate signal than temperature.

[1] Department of Forest and Conservation Sciences, University of British Columbia 3041-2424 Main Mall, Vancouver, BC V6T 1Z4, Canada. [2] Climate Research Division, Environment and Climate Change Canada 3800 Finnerty Rd, Victoria, BC V8P 5C2, Canada. Correspondence and requests for materials should be addressed to C.R.M. (email: c_mahony@alumni.ubc.ca)

Understanding climate change risks involves anticipating pressures on the adaptive capacity of socio-ecological systems. Given the complexity of these systems, it is important to identify general drivers of climate impacts that bridge the idiosyncratic responses of individual species, ecosystems, and societies. One prominent approach to assessing coarse-filter ecological and economic risks is to quantify climatic changes relative to local climatic variability[1]. The premise of this approach is that biological and human populations are locally adapted[2] to the year-to-year variability of their environments, and have coping mechanisms for climatic changes within this range of variability[3]. The hypothesis that adaptive capacity scales with environmental variability has some theoretical[4,5], experimental[6,7] and observational[7–9] support in the context of ecosystems, where environmental variability is a component of natural selection on the life history, demographics, population genetic variation, and phenotypic plasticity of organisms. Further, historical variability is the range of conditions in which cultural and scientific knowledge about ecosystems has developed[10]. Departures from historical variability not only increase the potential for ecological disruptions to food security and ecosystem services[11] but also represent locally unfamiliar conditions in which adaptive responses may not be apparent to human communities[12]. The magnitude of a departure from local interannual variability therefore is an important indicator of climate change risks.

In this paper, we use the term natural variability to encompass the pre-industrial historical variability of real-world climates as well as the non-anthropogenically forced internal variability of climate models. Departures from natural variability are commonly measured using the signal-to-noise ratio (S/N), which expresses changes in one climatic variable (the signal) relative to the scale of natural variability in that variable (the noise)[13]. Typically, the noise is defined as standard deviations of interannual variability (symbolized with sigma, $\sigma$), in which case S/N is equivalent to a standardized anomaly. S/N was originally used for the detection of climate change[14]. However, following the general logic of local adaptation to natural variability, S/N has become a widespread metric in many fields of climate change impact assessment, such as the human perceptibility of climate change[15,16], heat extremes[17], risks to natural[18] and agricultural[11] ecosystems, and general societal risk[12]. The timing of departures from natural variability has received particular attention, in variables including temperature[19–21], precipitation[22,23], biogeochemical cycles[24,25], sea level[26], and specific ecological drivers[27]. These time-of-emergence studies have identified regions of large and rapid departures from natural variability, notably the observed and projected warming of the tropics where human populations[12] and socioeconomic vulnerability[28] are concentrated. These coarse-scale studies assist the prioritization of more detailed regional climate risk assessments[29].

Risk assessments based on departures from natural variability have predominantly[30,31] analyzed individual climate variables. Interactions and dependencies between climate variables are increasingly recognized as important contributors to climate change impacts[32,33], and have implications for departures from natural variability. For example, summer precipitation (Pr) and mean daily maximum temperature (Tx) are negatively correlated over most land areas[34]. This correlation is driven primarily by evaporative cooling, but also by reflection of sunlight by clouds, and land-atmosphere coupling[35]. As a result of this relationship, natural variability in many terrestrial locations lies along an axis from warm-and-dry to cool-and-wet conditions, and excludes conditions that are simultaneously much warmer and wetter than average. A climate change trend perpendicular to this axis, toward warmer-wetter conditions, can produce a larger and earlier departure from natural variability than in either Tx or Pr alone

(Fig. 1). This type of climate change trajectory is projected by climate models to occur in many regions[36]. This example illustrates how extreme conditions can arise from unusual combinations of climate variables that are individually not in an extreme state. We use the term departure intensification to describe a multivariate climate change signal that is stronger relative to natural variability than the signals of its component variables. Our use of the term departure is synonymous with the use of emergence in the S/N and time-of-emergence literatures mentioned previously[22].

Like all climate departures[28], departure intensification in summer temperature and precipitation is a coarse-filter indicator of ecological risk. Nevertheless, some specific potential impacts can be identified. Host–parasite interactions can be much more disruptive to the host species than the physiological effects of environmental change alone[2]. Fungal and microbial plant pathogens are of particular concern because of their responsiveness to growing season temperature and precipitation[37], and their impacts on food security, forest health, and ecosystem services. For example, the global increase in the incidence and severity of *Dothistroma* fungal needle blight outbreaks in pine plantations has been linked to anomalously warm and wet conditions in many regions[38]. Mosquito-borne diseases such as malaria and dengue fever are also of concern, because both the vector and pathogen are responsive to heat accumulation and standing water availability[39]. The departure intensification effect suggests that the response of these agents to simultaneously warmer and wetter conditions may be out of proportion to their local historical response to unsynchronized anomalies of similar magnitude in temperature or precipitation alone. These disruptions could offset the benefits of increased precipitation in reducing hotter droughts[40] induced by regional

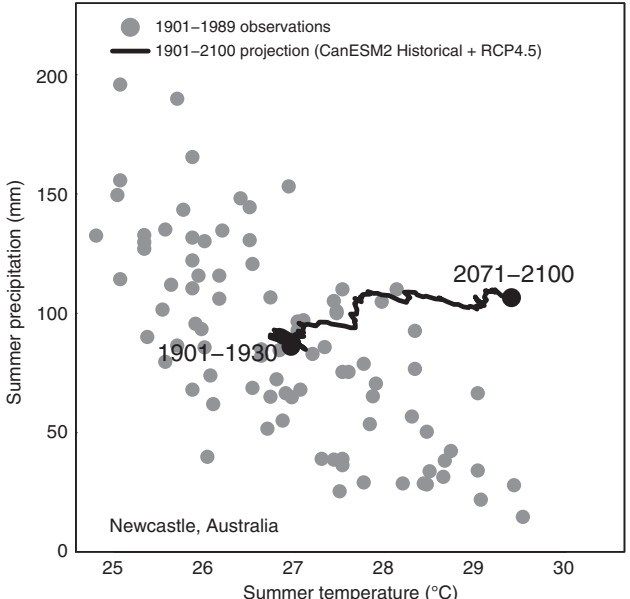

**Fig. 1** Projected departure from historical climatic variability in a correlated temperature-precipitation regime. Summer precipitation and mean daily maximum temperature at Newcastle, Australia are negatively correlated ($r = -0.7$) in historical observations (gray dots). As a result, a climate change trajectory into warmer-wetter conditions (black line) can depart from the joint historical variability of temperature and precipitation while remaining within the individual historical ranges of variability of both temperature and precipitation. Historical observations are obtained from the CRU TS3.23[44] source stations. The climate change projection is the 30-year running mean of the CanESM2 Historical + RCP4.5 ensemble mean projection

warming. Assessment of departure intensification in precipitation, temperature, and other coupled variables can provide an early warning of rapid ecological change that would not be apparent in analysis of individual climate variables.

The goal of this paper is to describe the phenomenon of climate departure intensification using the example of summer Tx and Pr. We use a multivariate S/N approach to examine intermodel variation in departure intensification in RCP4.5 projections from six Coupled Model Intercomparison Project Phase 5 (CMIP5[41]) global climate models for the 1850–2100 period. We demonstrate that departure intensification is contingent on the degree to which a bivariate climate change trajectory is misaligned with (orthogonal to) the dominant mode of interannual variability in coupled variables with correlations of $|r| > 0.5$. Station observations of summer Tx and Pr exhibit this level of correlation in distinct regions on all continents, narrowing the spatial extent for investigations of departure intensification. Our results suggest that precipitation trends can be important to the strength of multivariate climate change signals relative to natural variability, even though the S/N of precipitation variables are typically much less than those of temperature.

## Results

**Departure metrics**. We quantify the magnitude of climate departures as the proportion of years in the preceding 30-year period that are $>2\sigma$ (~1-in-20-year) anomalies of the internal natural variability of the model (i.e., historicalNat runs). We call this metric the $2\sigma$ proportion. By definition, assuming normality, the $2\sigma$ proportion has a null value of 0.046 in a stationary climate; i.e., the $2\sigma$ exceedance probability in the normal distribution. A non-stationary climate can be said to have fully departed from natural variability when the $2\sigma$ proportion is 1. We calculated the $2\sigma$ proportion for bivariate Tx-Pr anomalies using sigma dissimilarity[42], a parametric method of calculating multivariate S/N and standardized anomalies using Mahalanobis distance[43]. We use the term departure difference for the difference between the bivariate $2\sigma$ proportion and the larger of the $2\sigma$ proportions of Tx or Pr alone. Positive departure differences indicate that the bivariate S/N is stronger than either of the univariate S/N values individually. We use the maximum departure difference during the twenty-first century as our primary metric of departure intensification to allow comparisons between regions with various timings of departures.

**Factors contributing to departure intensification**. Departure intensification is dependent on two factors: the strength of correlation ($r$) between climate variables; and the orthogonality ($\theta$) of climate change (Fig. 2). The strength of the projected climate change signal is stronger in temperature than in precipitation in almost all grid cells (Supplementary Note 1). Nevertheless, the precipitation trend determines the alignment of climate change with interannual variability: large departure intensification occurs where the bivariate trajectory of climate change is orthogonal (perpendicular) to the dominant mode of interannual variability, as in Fig. 2b, e. Departure intensification does not occur if the Tx-Pr correlation is low (Fig. 2c) or if the trajectory of climate change is aligned with the dominant mode of variability, i.e., toward warmer-drier conditions (Fig. 2d), except where the Tx-Pr correlation is very high (Fig. 2f). Departure intensification can be underestimated by our metric where the climate change signal is very strong (Fig. 2g), as both the univariate and bivariate anomalies rapidly exceed a $2\sigma$ proportion of 1.

**Intermodel variation**. There is considerable intermodel variation in summer Tx-Pr correlations and the orthogonality of climate change (Supplementary Note 2), and as a result there is large intermodel variation in the magnitude and spatial pattern of departure intensification (Fig. 3). Nevertheless, departure intensification is evident in all models. On average, 23% of the land area of each model has a maximum departure difference >0.2 (i.e., >6 more years of $2\sigma$ anomalies in the preceding 30 years). The IPSL-CM5A_LR and CanESM2 models represent the intermodel range of departure intensification, with maximum departure differences >0.2 on 9% and 34% of their land area, respectively. Low departure intensification in IPSL-CM5A_LR is due to its low correlation between summertime Tx and Pr relative to the other models (Supplementary Note 2). The reduced intensification in the GISS-E2-R projection is likely due to its pronounced lower climate sensitivity (Supplementary Table 1).

The S/N of summer Tx-Pr climate change is highest in the tropics (Supplementary Note 3). However, departure intensification of Tx-Pr regimes is evident at most latitudes (Fig. 3). It is generally absent from the Boreal and Arctic regions due to low Tx-Pr correlations. There is a strong relationship between the magnitude of departure intensification and the relative timing of departure of the bivariate Tx-Pr climate signal. As suggested by Fig. 2b–g, the bivariate climate departure can occur several decades prior to the departure of Tx alone (Supplementary Note 3). On average, 16% (intermodel range of 7–23%) of the land area of each model departs from the joint variability of Tx and Pr at least 10 years prior to the departure of either Tx or Pr alone (Supplementary Note 3). These departure timings are based on a threshold $2\sigma$ proportion of 0.25, which represents a 10-fold increase in the frequency of warm $2\sigma$ anomalies. The role of departure intensification in the relative timing of bivariate and univariate climate departures is strongly influenced by how climate departure is (arbitrarily) defined. Consequently, the relative timing of departures is a less reliable indicator of departure intensification than the maximum departure difference.

**Minimum variable correlation for departure intensification**. Bivariate climate departures are detected in tropical and subtropical regions of most climate models by the year 2020 (Fig. 2 and Supplementary Note 3). This suggests that departure intensification may be detectable in the observational record at low latitudes, consistent with other studies that have detected emergence of a local warming signal[3]. In extratropical regions where the climate signal has not yet emerged from natural variability, departure intensification likely is not yet detectable in the observational record. Nevertheless, all models indicate that regardless of the orthogonality of the bivariate climate change trajectory, substantial departure intensification only occurs where there is a strong negative correlation ($r < -0.5$) between summertime Tx and Pr (Fig. 4). This general relationship provides a rule of thumb for evaluating the observational record for regions that are susceptible to intensified climate departures. CRU TS3.23 precipitation stations[44] exhibit strong negative summertime Tx-Pr correlations in several distinct regions on all continents, particularly in China, central Asia, Australia, central USA, western Canada, Argentina, southern Africa, and the African Sahel (Fig. 5). These regions are a priority for detection of departure intensification in the observational record, determining the probable trajectories of precipitation change, and identifying specific local impacts of departure intensification.

## Discussion

We have demonstrated that a multivariate climate change trend can be stronger, relative to natural variability, than all of the

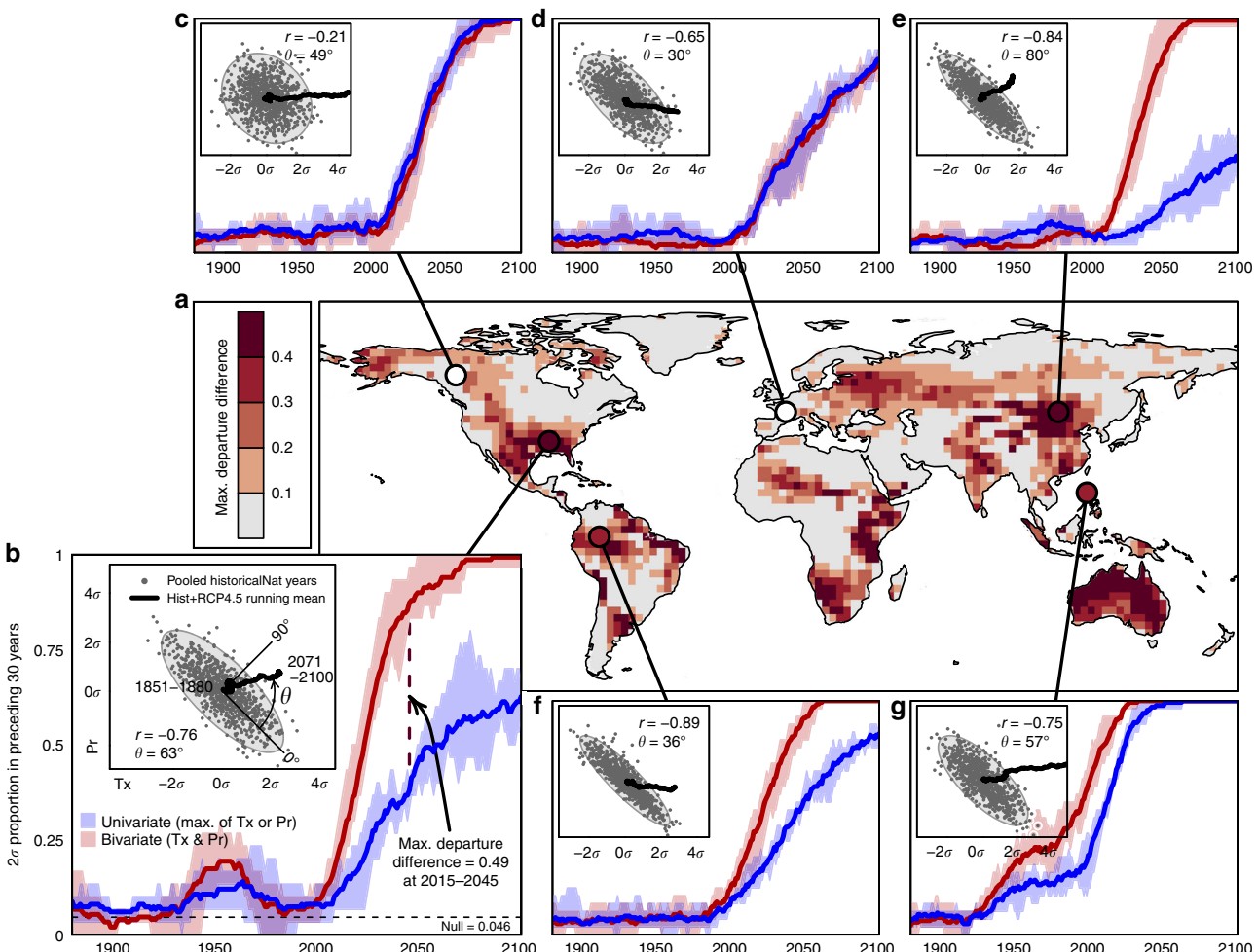

**Fig. 2** Univariate and bivariate climate departures for summer precipitation and mean daily maximum temperature at selected locations; CanESM2 RCP4.5 ensemble projection. **a** Terrestrial regions where bivariate departures from natural variability exceed univariate departures, measured as maximum departure difference during the 1880–2100 period. **b** Time series of univariate and bivariate $2\sigma$ proportions near Memphis, USA. Univariate values are the larger of the precipitation (Pr) and mean daily maximum temperature (Tx) $2\sigma$ proportions. Bivariate proportions are assessed against the joint distribution of Tx and Pr. Shaded regions of the time series indicate the spread of the 5-run CanESM2 ensemble; solid lines are the ensemble mean. **b** (Inset) Scatterplot of CanESM2 historicalNat projections of Tx and Pr, pooled and normalized across five runs of the 1850–2006 period. Correlation (Pearson's $r$) is measured using the normalized values of Tx and Pr. The orthogonality of climate change ($\theta$) is the angular displacement of the bivariate climate change trajectory from the dominant axis of bivariate interannual variability. **c**–**g** Equivalent plots for **c** Whitehorse, Canada; **d** Central France; **e** Eastern Mongolia; **f** the Colombian Amazon; and **g** Manilla, Philippines. The dashed line in plot **b** indicates the expected $2\sigma$ proportion of 0.046 in a stationary climate with Gaussian (normal) variability

individual trends of its component variables. This departure intensification effect occurs when the climate change trajectory is misaligned with the dominant mode of interannual variability in correlated climate variables. In the case of summer Tx and Pr— the focus of this study—these conditions hinge on stable or increasing precipitation. We found considerable intermodel variation in the amount and spatial pattern of departure intensification, due largely to intermodel variation in projected regional precipitation trends and the strength of simulated Tx-Pr correlations. Nevertheless, we also found that pronounced departure intensification is consistently limited to simulated climates with a summer Tx-Pr correlation stronger than −0.5. This result provides considerable direction for prioritizing regions for further risk assessments based on observed correlations.

The climate variables in this study—summer Tx and Pr— reflect our primary goal of demonstrating departure intensification at a global scale. We selected the Boreal and Austral summer (JJA and DJF, respectively) to approximate the warm season, and

to be consistent with previous studies of temperature-precipitation coupling[34,35] that are foundational to our study. However, we acknowledge that this season selection is problematic in the tropics and some subtropical regions, and have provided a parallel analysis using an alternative definition of summer—the hottest three consecutive months in each grid cell (Supplementary Note 4). We selected Tx as our temperature element in recognition that daily mean temperature could conflate different and potentially opposing physical processes driving daytime vs. nighttime temperature-precipitation correlations in many regions[35]. We acknowledge that daily mean or minimum temperatures may be more salient elements for some regions and ecological processes. Identification of departure intensification in other economically and ecologically important coupled climate variables is a priority for future research. Our finding that departure intensification is limited to variability correlations stronger than −0.5 provides a guideline for interrogating the observational record for other coupled variables that are subject to departure intensification.

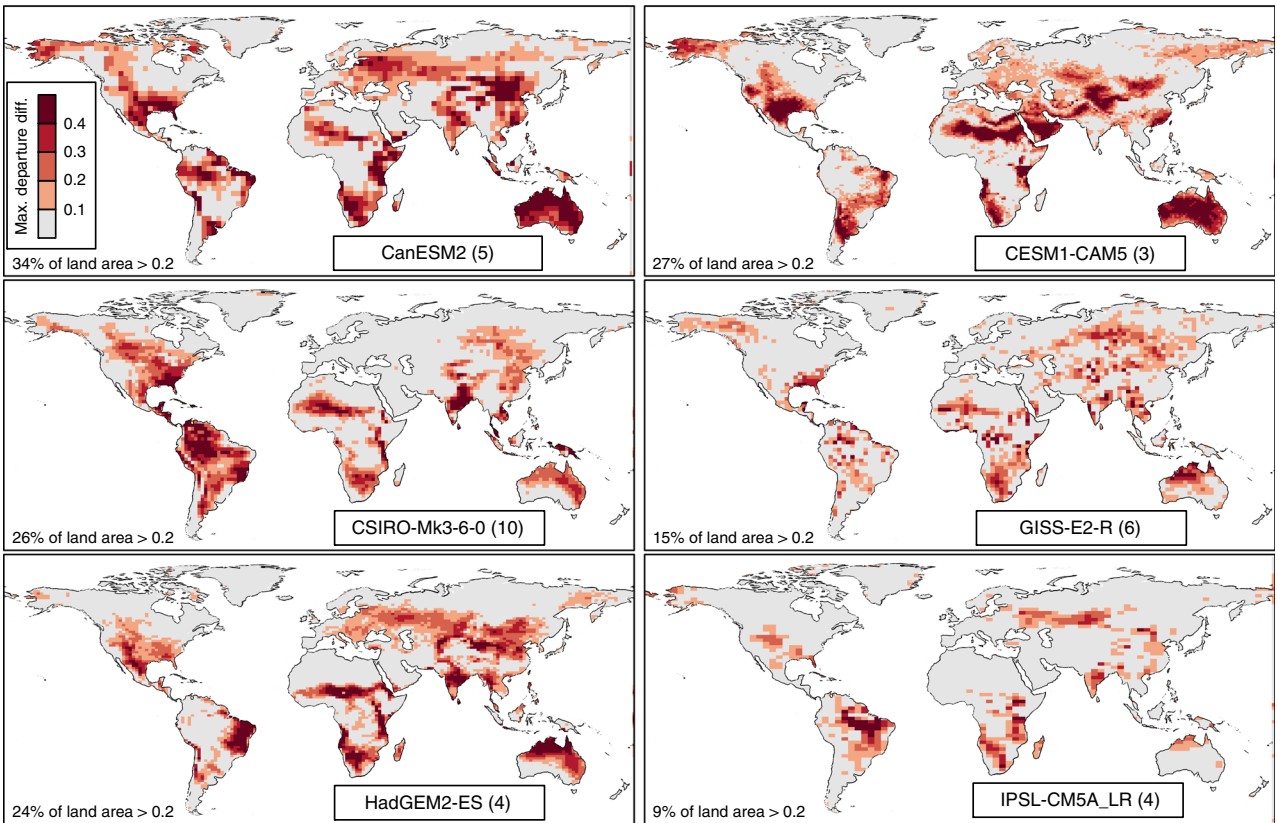

**Fig. 3** Intermodel variation in departure intensification in summertime temperature and precipitation. Departure intensification, measured as maximum departure difference, is calculated from the mean time series of univariate and bivariate $2\sigma$ proportions of historical + RCP4.5 runs for each CMIP5 model. The number of runs in each ensemble is indicated in parentheses next to the model name. Departure difference is the frequency of bivariate $2\sigma$ proportions minus the frequency of univariate (greater of temperature or precipitation) $2\sigma$ proportions, both with respect to the 1850–2005 historicalNat simulations in each model. Positive values indicate a greater frequency of departures in the bivariate temperature-precipitation regime than in temperature or precipitation alone. Summer months are JJA (Northern Hemisphere) and DJF (Southern Hemisphere)

The impacts of climate departures are subject to the timescales over which maladaptation to unfamiliar local conditions is mitigated by gene flow, innovation, and other non-disruptive adaptive processes. In contrast to studies using a recent reference period[20,22,28], our use of a pre-industrial baseline ignores local adaptation that has occurred during the industrial period, and may overstate the timing and magnitude of some of the disruptions associated with climate departures. The timescales of local adaptation are an important consideration in the assessment of the specific impacts of climate departures.

We have shown that precipitation can have an important role in climate departures at interannual timescales, even though precipitation signals themselves are generally not projected to emerge from interannual climatic variability at the local scale[23]. CMIP5 models generally agree on the direction of twenty-first-century regional precipitation trends but differ substantially in magnitude, not only due to structural differences among climate models[45] but also due to the internal variability of each model run[46]. The sensitivity of departure intensification to precipitation trends highlights the importance of precipitation as a source of uncertainty in projections of ecological responses to climate change.

Departure intensification is a decadal-scale compound event, a class of climate extremes that arise from interactions and dependencies among multiple climate variables[32]. The case of departure intensification illustrates that some compound events are primarily multivariate and can remain undetected in univariate indices that synthesize multiple climate variables, such as the standardized precipitation-evapotranspiration index[47] and wet-bulb temperature[48]. The World Climate Research Programme has established compound events as a research priority[49], and techniques for identifying multivariate climate extremes are emerging[31]. However, investigations of compound events are sparse in many fields of climate change detection, impacts, and adaptation. Our study demonstrates a form of compound extremes arising from historically unusual combinations of conditions. These compound anomalies can occur both as single-year events and long-term climatic shifts. Identifying ecological and agricultural impacts associated with compound events and compound climate departures is an important area for further research.

Departures from natural climatic variability are a challenge to locally adapted natural and agricultural ecosystems. Decoupling of the climate change trend from the dominant historical mode of interannual variability can accelerate the rate at which locally unfamiliar climates develop, which may be a limitation on the ability of some organisms and societies to adapt to climate change[5–11]. Our study has focused on this effect in terrestrial summer temperature and precipitation. However, the cause of departure intensification—a climate change trajectory that is misaligned with natural variability—likely applies to other climatic drivers of ecosystem function. The potential for climate departures to be amplified in coupled climate variables is an important consideration for climate change risk assessments and adaptation planning.

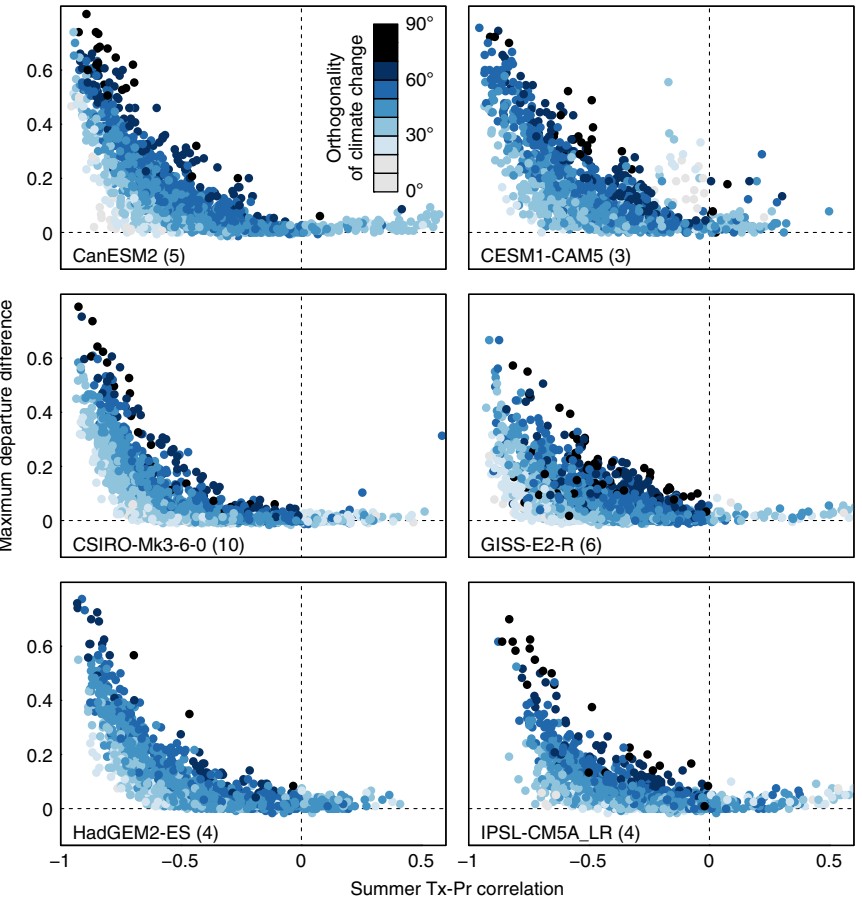

**Fig. 4** Relationship of departure intensification to temperature-precipitation correlations and the orthogonality of climate change in six CMIP5 models. Orthogonality increases departure intensification at any given correlation. However, substantial departure intensification is only present at temperature-precipitation correlations stronger than $r = -0.5$. Departure intensification, measured as maximum departure difference, is calculated from the mean time series of univariate and bivariate $2\sigma$ proportions of historical + RCP4.5 runs for each CMIP5 model. The correlation between summer mean daily maximum temperature (Tx) and precipitation (Pr) in each CMIP5 model is calculated from the r1i1p1 historicalNat run, typically for the years 1850–2006. The orthogonality of climate change ($\theta$) is the angular displacement of the RCP4.5 single-model ensemble mean projected normals for the 2051–2100 period from the dominant axis of bivariate interannual variability. The number of runs in each ensemble is indicated in parentheses next to the model name. Oceans and Antarctica are not plotted. A random sample of $n = 1500$ grid cells is plotted for each model to equalize plotting density. Dashed lines indicate a value of zero in the x- and y-axis

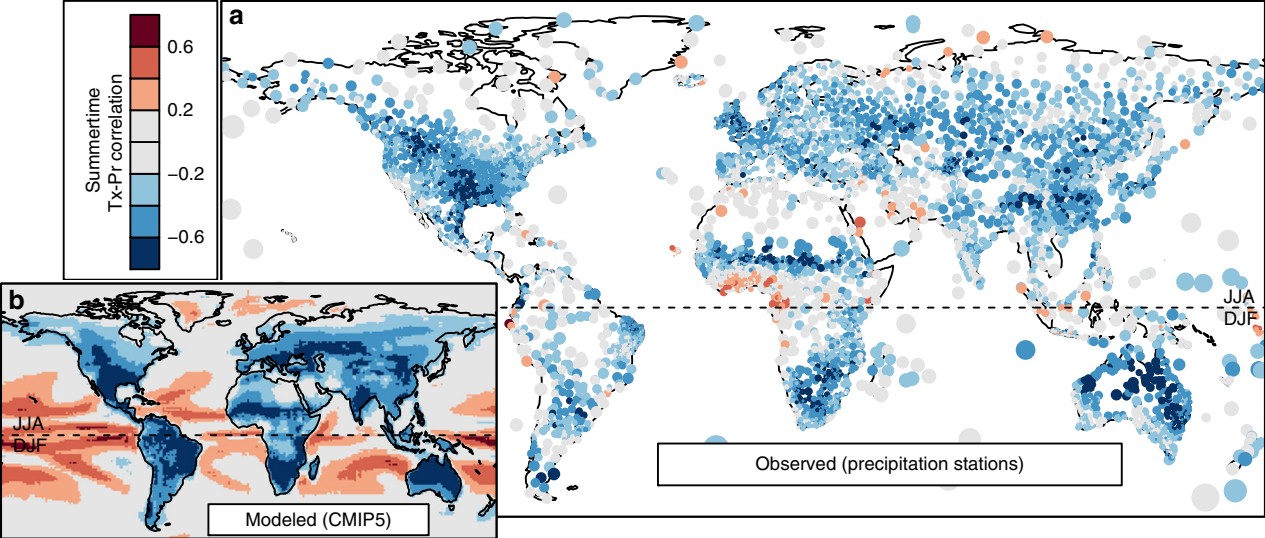

**Fig. 5** Observed and modeled correlations between summer precipitation and mean daily maximum temperature. Spatial patterns in the correlation between mean daily maximum temperature (Tx) and precipitation (Pr) of the Boreal (JJA) and Austral (DJF) summer are generally consistent between **a** CRU TS3.23[44] precipitation stations and **b** the six-model ensemble mean of CMIP5 historicalNat simulations. To reduce overplotting, point size of precipitation stations is proportional to distance from the third nearest station

## Methods

**Reference interannual climatic variability**. We used historical natural forcings (historicalNat) model runs as reference variability for each CMIP5 model. HistoricalNat runs exclude anthropogenic forcings such as greenhouse gas emissions and land use change over the period 1850–2005, but include historical natural

radiative forcings (e.g., solar cycles and volcanic eruptions)[41]. Pre-industrial control (piControl) runs are an alternative source of reference variability, but may have slightly biased natural variability due to the absence of volcanic forcings. For each model, we pooled all of the historicalNat runs available (excluding perturbed physics runs), such that the reference variability encompasses several historicalNat realizations of the 1850–2005 period. For example, the CanESM2 model, with 5 historicalNat runs, has $n = 5$ runs × 156 years/run = 780 years of reference variability. This large sample size facilitates reliable characterization of the tails of the distribution of reference variability and minimizes overestimation of anomalies outside of the reference period[50] (Supplementary Note 5).

**Univariate standardized anomalies**. Climate variables (e.g., with units of °C or mm) can be expressed as standardized anomalies, with units of standard deviations (symbolized as $\sigma$) of interannual variability over a multidecadal reference period. Standardized anomalies are traditionally calculated as $z$-scores, by subtracting the reference period mean and dividing by the reference period standard deviation[51]. However, climate variables are often not normally distributed (e.g., substantial skewness), with the result that $z$-scores can overestimate the probability of an anomaly on one tail of the distribution, and conversely underestimate it on the other. To overcome this problem, we normalized univariate standardized anomalies of both temperature and precipitation using a form of non-parametric quantile mapping[52], which additionally preserves the magnitude of the climate change relative to reference variability. We determined that our analyses are not substantially different under other methods of correcting for non-normality, including univariate parametric quantile mapping (as used in the standardized precipitation index[53]), bivariate kernel density estimation, and multivariate quantile mapping[54] (Supplementary Note 6). The reason for this robustness is that as a binary metric, the $2\sigma$ proportion (see below) is unaffected by the effect of normalization on the magnitude of anomalies beyond the $2\sigma$ threshold.

**Bivariate standardized anomalies**. We used sigma dissimilarity[42] to measure deviations from the bivariate distribution of temperature and precipitation. First, absolute bivariate anomalies are calculated as Mahalanobis distances from the centroid (multivariate mean) of the reference variability. Then, the squared distances are converted to percentiles of the $\chi^2$ distribution, which can be expressed as sigma levels (i.e., $1\sigma$, $2\sigma$, and $3\sigma$ for ~68th, 95th, and 99.7th percentiles, respectively). Sigma dissimilarity facilitates direct probabilistic comparison between univariate and bivariate anomalies (Fig. 6). Orthogonality of climate change ($\theta$) is measured as the inverse tangent of the relative magnitude of the 2051–2100 normals ($\Delta$) of the principal components (PC1 and PC2) of reference variability, providing degrees of angular displacement from the dominant axis of interannual variability: $\theta = \arctan(\Delta PC2/\Delta PC1)$.

**Climate departure metrics**. The anomalies of projected twenty-first century climate change commonly exceed $3\sigma$ (an ~$1/(1 − 0.9973)$ = 1-in-370-year exceedance), and therefore exceed the limits of statistical inference from even very large samples of modeled reference variability. This precludes quantification of the mean anomaly as a measure of departure from natural variability. Instead, we quantify

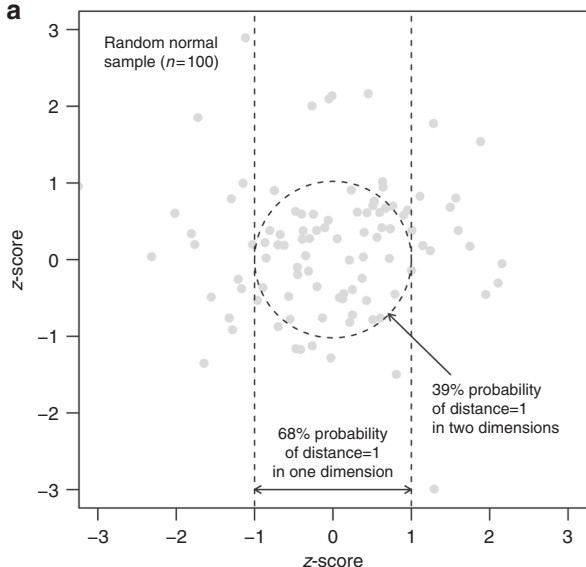

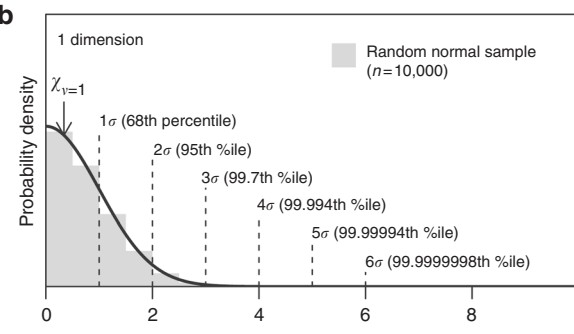

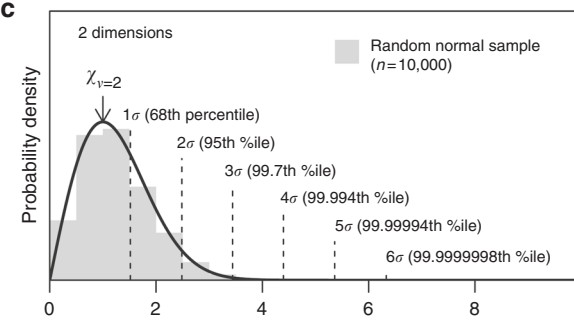

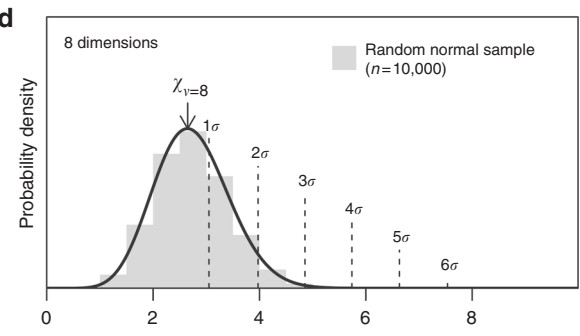

**Fig. 6** Illustration of sigma dissimilarity for direct probabilistic comparison between anomalies of different dimensionality. **a** The effect of dimensionality on the probabilities of distances can be visualized using a sample from a standard bivariate normal distribution. In any one dimension, there is a 68% probability that an observation will be within one standard deviation ($z$-score) of the mean. In two dimensions, the probability that an observation will fall within a distance of one from the centroid is reduced to 39%. **b**–**d** As dimensionality increases, the mean distance to observations increases, and even the observation closest to the mean becomes distant. This effect of dimensionality on the probability density of distances in multivariate normal data is described by the chi distribution ($\chi$) with degrees of freedom ($\nu$) equaling the dimensionality of the data[51]. The sigma dissimilarity metric is a re-expression of chi percentiles using the terminology of univariate $z$-scores; i.e., 1, 2, and 3 sigma ($\sigma$) for the 68th, 95th, and 99.7th normal percentiles, respectively. The chi distribution in one dimension is a half-normal distribution, and the sigma levels correspond to distance. This result is expected because Mahalanobis distances in one dimension are the absolute values of $z$-scores. At increasing dimensionality, the sigma levels shift away from the origin. For example, $2\sigma$ (the 68th percentile) occurs at Mahalanobis distances of 2, 2.5, and 4.0 in one, two, and eight dimensions, respectively. By extending sigma levels into multiple dimensions, sigma dissimilarity serves as a multivariate $z$-score

the magnitude of climate change as the proportion of years in the preceding 30-year period that are $>2\sigma$ (~1-in-20-year) anomalies relative to the distribution of reference variability. We call this metric the $2\sigma$ proportion. We define the term departure difference as the difference between the bivariate and univariate $2\sigma$ proportions. Univariate $2\sigma$ proportions are calculated as the maximum of the $2\sigma$ proportions of temperature and precipitation. As a noise-reduction measure, we calculated the departure difference for each year from the mean of the $2\sigma$ proportions of several historical + RCP4.5 runs. The null values of departure difference are slightly negative (Supplementary Note 7): where the correlation is non-significant and there is no trend in precipitation, the bivariate $2\sigma$ proportion is less than the univariate $2\sigma$ proportion by up to 0.1 depending on the magnitude of the temperature change. Positive departure differences indicate that the bivariate S/N is stronger than the univariate climate S/N. Pseudocode for calculation of maximum departure difference is provided in Supplementary Note 8.

**Model selection**. To provide a robust representation of natural variability and the anthropogenic climate change signal, we selected CMIP5 climate models with at least three historicalNat runs and three RCP4.5 runs (Supplementary Table 1). We excluded CCSM4, which is superseded by CESM1-CAM5[55]. We did not exclude models on the basis of model diagnostics.

**Observed temperature-precipitation correlations**. To complement previous analyses of gridded variability, we assessed observed Pr-Tx correlations at the station level in order to obtain local-scale estimates of interannual variability that are relatively free from grid-box averaging artifacts[56]. We anchored our analysis on precipitation stations because spatial correlation of precipitation is much lower than in temperature[57]. We obtained CRU TS3.23[44] precipitation station observations from the source station files. We excluded stations with <30 years of summer precipitation totals in the 1901–2013 period. Since precipitation and temperature are often not assessed at the same station, we used the CRU TS4.0 gridded Tx time series at the location of each precipitation station.

**Software**. All analyses and results were produced in R[58]. Coastlines were mapped with spatial data provided in the R package "rworldmap"[59].

**Data availability**. The model output analyzed here is publically available via the CMIP5 archive. R code and sample data sufficient to reproduce this study are available within the Supplementary Information files associated with the online version of this manuscript.

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

## Acknowledgements

We acknowledge the World Climate Research Programme's Working Group on Coupled Modelling, which is responsible for CMIP, and we thank the climate modelling groups for producing and making available their model output. We thank Dr. Sally Aitken, Dr. Amy Angert, Megan Bontrager, Dr. Reza Najafi, Hui Wan, and Dr. Bárbara Tencer for providing helpful comments on drafts of this paper. Funding for this research was provided by an NSERC PG fellowship to C.R.M. and an NSERC Discovery Grant to Dr. Sally Aitken.

## Author contributions

Both authors set the study goals. C.R.M. designed the study and conducted data analysis under the guidance of A.J.C.; C.R.M. wrote the paper and both authors contributed to the editing process.

## Additional information

**Competing interests:** The authors declare no competing financial interests.

