## [Peer Review File · Nature Communications]

Reviewers' comments:

Reviewer #1 (Remarks to the Author):

Review of "Wetter summers can intensify departures from natural variability in a warming climate." by Mahony and Cannon.

This study applies a signal-to-noise approach to maximum temperature and precipitation in combination, and identify where this approach finds earlier climate change influences compared with a univariate methodology.

The results are interesting and the paper is very well-written and will make an excellent contribution to the literature. I only have a few minor comments.

Comments:

L34: Most time of emergence studies use a recent baseline (e.g. Hawkins and Sutton 2012) so it's worth noting that this study is looking at the human-induced signal in isolation (similar to early base periods used by Mahlstein et al. (2012) and King et al. (2015)).

Figure 1a: I would suggest plotting the equator in the background with JJA to the north and DJF to the south to highlight that you are studying the summer season.

Figure 1: There are interesting differences between the observed and simulated TX-Pr correlations such as equatorial West Africa, as well as areas where the correlations are remarkably similar, like Australia and the US. It would be worth commenting on potential causes of differences between the observed and simulated correlations with respect to West Africa- are there individual models with similar patterns? Do we trust the observations in this region?

L55: While I like the multivariate approach here, I suspect that the TX change is dominating over the Pr change in line with the findings of previous studies.

Figure 2b: I think the label should read "Historical + RCP4.5 30-yr running mean"

L89: Is the Tx change always larger than the Pr change?

Figure 3: I'm a bit confused as to exactly what is being shown here- is it the average departure between runs of the same model? Is the average a median or a mean (a mean-average is sensitive to outliers)? It would be useful to have an indication of the level of agreement between land area proportions between runs of the same model and different models (similar to King et al. 2015; Bador et al. 2016).

L124-127: This result contrasts with other studies which use an early baseline and already find indications of emergence (e.g. Mahlstein et al. 2012; King et al. 2015). A comment on this difference would be useful.

References:

Bador M, Terray L and Boé J 2016 Emergence of human influence on summer record-breaking temperatures over Europe Geophys. Res. Lett. 43 404–12

Hawkins E and Sutton R 2012 Time of emergence of climate signals Geophys. Res. Lett. 39 L01702

King A D, Donat M G, Fischer E M, Hawkins E, Alexander L V, Karoly D J, Dittus A J, Lewis S C and

Perkins S E 2015 The timing of anthropogenic emergence in simulated climate extremes Environ. Res. Lett. 10 094015

Mahlstein I, Hegerl G and Solomon S 2012a Emerging local warming signals in observational data Geophys. Res. Lett. 39 L21711

Reviewer #2 (Remarks to the Author):

Wetter summer can intensify departures from natural variability in a warming climate
by Colin R. Mahony and Alex J. Cannon

Reviewer: Milan Flach, Max Planck Institute for Biogeochemistry

*Suggestion for the Editor: *

The paper 'Wetter summer can intensify departures from natural variability in a warming climate' by Colin R. Mahony and Alex J. Cannon is well presented, well written, timely, and important.

The authors show that dependencies in earth system models between multiple variables (here: temperature and precipitation) cannot be neglected when computing the 'time of emergence' or departure from natural variability at each model location. Specifically they show, that many regions on Earth depart faster from natural variability in case one considers the multivariate correlations structure of temperature and precipitation. They provide sufficient evidence for their conclusions.

Technically, they use the Mahalanobis distance to the mean as a measure of multivariate departure from natural variability. The statistical analysis via Mahalanobis distance is sound as such. Although the technique dates back several decades ago and is common in other fields of science, it is not commonly used in this particular field of science to the best of my knowledge. Thus, the technique is novel for this particular application on CMIP5 models. The novel multivariate perspective furthermore offers advance to previous journal articles on this topic. In general, the manuscript allows to reproduce the results. There are only minor issues to improve understandability, which are described in detail below.

The paper is timely and interesting to others in the field as it goes in line with recent activity on multivariate extremes / compound events. I believe, that it sets new standards for future publications on the time of emergence / departure time, as it highlights the importance of the multivariate perspective on this particular topic.

I would recommend to accept the paper for publication in nature communications after the authors carefully addressed the following questions.

Best regards,
Milan Flach

*Major Comments to the Authors: *

1.) The definition of summer in the paper seems to be a bit too simplistic to me. Specifically, as already mentioned in the discussion (p.6, l.158), it hinders a straightforward interpretation of your results in the tropics (p.5, l.110). I was wondering, if it would be possible to change the definition of your summer (the warm period) to be the hottest three month of tmax (comparable e.g. to (1)). This rather small change would make your statement much stronger in low latitude regions, which are very much affected by the intensification of departure.

2.) You state in the introduction, that the environment is locally adapted to the climate in which it is living (p.1, paragraph 1/2). You are using runs of the CMIP5 models with historical natural variability to quantify the normal local variability within each gridcell. In case, one uses historical runs, one assumes that no further adaptation is taking place, which is not very plausible to my mind. Nevertheless, your approach is valid and commonly used. However, I was wondering about the time scale of local adaptation to climate. May be you could elaborate on this a bit further in the introduction. Defining a specific time scale for adaptation to the local climate would be another option for such kind of studies, which goes behind the scope of this paper, but nevertheless should be mentioned.

3.) On p.4, l 95, you are stating that large departure correlation goes in line with orthogonality to the natural mode of interannual variability. Would it be possible to provide further (quantitative) evidence for this finding as supplementary material? Probably, the second component of a principal component analysis (PCA) could serve as a measure of orthogonality to natural variability. PCA is technically very much related to / the same as the Mahalanobis distance to the mean.

4.) On p. 7, ll. 224-225 you are defining the univariate 2 sigma proportion as the maximum of the 2 sigma proportions of temperature and precipitation for computing the departure difference. May be it would improve understanding, if you add a (pseudo) formula here, like $\max(\text{prop}(Tx), \text{prop}(Pr))$, in case I understood it correctly. It would also be nice if you elaborate further on the reason for this specific choice.

Furthermore other choices would be thinkable, like computing the 2 sigma proportion as sigma exceedances of either precipitation or temperature ($\text{prop}(Tx \text{ or } Pr)$), or computing the 2 sigma proportion of the marginal distributions of temperature or precipitation separately ($\text{prop}(Tx)$ or $\text{prop}(Pr)$). The latter choice would probably be very helpful to disentangle / attribute high departure differences to the univariate drivers. Thus, I would highly appreciate if you could provide some supplementary material on e.g. the latter alternative choice.

Minor Comments to the Authors:

p.1, l. 3.: 'interactions between climate variables'. The meaning of the word 'interactions' goes much further, than what you are actually looking at. I would favour a more neutral wording like 'correlations' or 'dependencies'. (similar: p.4, l. 48; p. 5, l. 146; p.6, l. 173, 182, 184, 187)

p.4, l. 87: 'sigma dissimilarity, a novel method.' The technique of 'sigma dissimilarity' translates anomaly scores of the multivariate Mahalanobis distance into percentiles of the theoretically received chi-squared distribution. Although I am in favour of the 'sigma dissimilarity' technique, calling this method 'novel' without a detailed literature review seems a bit exaggerated to me. The Mahalanobis distance is used since 1936 (2) and used for novelty / anomaly detection as Hotelling's T^2 control chart since 1947 (3). The reviewer would be wondering, if nobody used percentiles of the chi-squared distribution in these decades. The reviewer would like to ask the authors to remove 'novel' from the sentence or to reword and clarify: what is particularly novel on the sigma dissimilarity technique?

p. 5, ll.132-134: This finding is particularly interesting for estimating departure intensification in (methodologically more simplistic) impact focussed studies on novel climates. However, whether higher correlations lead to a higher 'risk' in terms of impact on humans or biological systems is questionable / not a direct result of this study. It would require further research or might be better moved to the discussion part.

p. 6, l. 175: Did you actually try to detect the multivariate events by SPEI?

p. 6, ll. 178-179 and 184: Misleading for my understanding is the 'mechanism by which compound events to occur', which you refer to. As far as I understood the study, it is focussed on statistical correlations between precipitation and temperature and not on 'mechanisms' which would induce

those kind of departures. Could you please elaborate further on what they specifically mean or rephrase the statement.

p. 6, ll. 182-183: 'We have demonstrated that variable interactions can accelerate the rate at which locally unfamiliar climates develop, which is a critical limitation on the ability of organisms and societies to adapt to climate change.' The second part of this statement is not a direct finding of this study and requires a citation as such.

p. 6, l.190: Mentioning the reference period immediately rises the question: which reference period? It would probably be helpful for better understanding to include a reference to the later section on this topic.

p.7, l. 204: Please add a citation for the Mahalanobis distance ,e.g. (2) or (3) below, and/or reference number 31 from your manuscript (as more recent one).

p. 7 / 8 / 9: References:

- 8. check page numbers
- 23. check page numbers / add DOI
- 31. discussion paper meanwhile published
- 33. check page numbers / add DOI
- 41. check page number and year
- 43. check page number

References:

(1) Zscheischler, J., & Seneviratne, S. I. (2017). Dependence of drivers affects risks associated with compound events. *Science Advances*, 3, e1700263)

(2) P. Mahalanobis, "On the generalised distance in statistics (vol.2, pp.49–55)," *Proceedings National Institute of Science, India*.

(3) Hotelling, H.: *Multivariate Quality Control - Illustrated by the Air Testing of Sample Bombsights*, in: *Techniques of Statistical Analysis*, edited by Eisenhart, C., Hastay, M. W., and Wallis, W. A., pp. 111–184, McGraw-Hill, New York, 1947.

Reviewer #3 (Remarks to the Author):

General Comments:

This is a very nice paper that clearly and thoroughly makes the case for considering pertinent variables in concert rather than in isolation when looking at changes in climate extremes. The authors focus on the relationship between summertime mean daily maximum temperature (Tx) and total precipitation (Pr), demonstrating that in many regions of the globe these variables are negatively correlated, with wet summers tending to be cool and dry summers tending to be hot. A warming climate, however, is pushing some regions towards warmer and wetter conditions. At these locations, the authors demonstrate that future conditions are likely to move beyond the historically-covered regions of the Tx-Pr parameter space more quickly than a climate change signal would emerge when considering either variable by itself. The authors made a good case for why these changes could have important consequences for many ecosystems. I believe this is a very strong paper and that *Nature Communications* is an excellent outlet for the material. I have listed a few minor suggestions for improvements below.

Specific Comments:

1. The introduction does a nice job of establishing the motivation for this study. You could draw

further support by referring to examples of studies of future climate extremes that used variables depending on both temperature and humidity, e.g. wet-bulb temperature (Pal and Eltahir, 2016; Im, Pal and Eltahir, 2017) or wet-bulb globe temperature (Knutson and Ploshay, 2016).

- Pal, J.S. and E.A.B. Elthair: Future temperature in southwest Asia projected to exceed a threshold for human adaptability. *Nature Climate Change*, 6, 197–200, doi:10.1038/nclimate2833, 2016.
- Im, Pal, and Eltahir: Deadly heat waves projected in the densely populated agricultural regions of South Asia. *Science Advances*, 02 Aug 2017; Vol. 3, no. 8, e1603322; DOI: 10.1126/sciadv.1603322.
- Knutson, T. R. and J. J. Ploshay: Detection of anthropogenic influence on a summertime heat stress index. *Climatic Change*, 138, doi:10.1007/s10584-016-1708-z , 2016.

2. Figure 1: “historicalNat” hasn’t been defined yet. I think it is only defined in the methods. You should mention that Figure S7 shows results from the individual models.

3. Figure 2: The specific examples of different possibilities (low Tx-Pr correlation, etc) are great, and the text describing Figure 2 very clearly explains the three relevant factors for departure intensification. The caption should mention the time period captured by the map in plot a. What does negative number of years for the red lines mean? According to the y-axis label they are a number of years (out of 30) divided by 30. If this is not correct, please clarify. In the methods you describe how the departure difference can be negative, but not how the 2σ proportion can be negative.

4. Line 164: Your results only show variability correlations stronger than -0.5, not +/-0.5. I understand that you are leading into talking about other variable pairs, but perhaps you can rephrase this sentence so it is cleaner.

5. Line 219: I think you should more explicitly indicate how you compute that a 3σ exceedance is a 1-in-370-year event.

The supplementary materials provide details that may be of interest to certain readers and enhance the rigor and thoroughness of the study. I believe they are correctly placed in the supplement.

Response to reviews

We would like to thank the reviewers for providing thorough reviews with helpful suggestions. We have described the changes we have made and any responses to comments below in blue font.

With the exception of some minor edits, the only change we have made that is unrelated to the reviewer comments is the deletion of the last two sentences of the abstract, to conform to the 150-word limit in the Nature Communications formatting standard.

Reviewers' comments:

Reviewer #1 (Remarks to the Author):

Review of "Wetter summers can intensify departures from natural variability in a warming climate." by Mahony and Cannon.

This study applies a signal-to-noise approach to maximum temperature and precipitation in combination, and identify where this approach finds earlier climate change influences compared with a univariate methodology.

The results are interesting and the paper is very well-written and will make an excellent contribution to the literature. I only have a few minor comments.

Comments:

L34: Most time of emergence studies use a recent baseline (e.g. Hawkins and Sutton 2012) so it's worth noting that this study is looking at the human-induced signal in isolation (similar to early base periods used by Mahlstein et al. (2012) and King et al. (2015)).

We have added a discussion paragraph that raises this issue.

Figure 1a: I would suggest plotting the equator in the background with JJA to the north and DJF to the south to highlight that you are studying the summer season.

Done

Figure 1: There are interesting differences between the observed and simulated TX-Pr correlations such as equatorial West Africa, as well as areas where the correlations are remarkably similar, like Australia and the US. It would be worth commenting on potential causes of differences between the observed and simulated correlations with respect to West Africa- are there individual models with similar patterns? Do we trust the observations in this region?

The CESM-CAM5 ensemble produces a similar but weaker pattern of positive correlations in western equatorial Africa (Figure S9). Given the scope and focus of the study, we cannot comment on the reasons for this difference or on the credibility of the observations in this region. However, Berg et al. (2015) speculate on physical processes responsible for model differences in simulated temperature-precipitation correlations. In the context of negative correlations, they found that an analysis of "five climate models, which were integrated with prescribed (noninteractive) and with interactive soil moisture over the period 1950–2100. [...] confirm the interpretation that negative correlations between seasonal temperature and precipitation arise through the direct control of soil moisture on surface heat flux partitioning, the presence of widespread negative correlations when soil moisture–atmosphere interactions are artificially removed in at least two out of five models suggests that atmospheric processes, in addition to land surface processes, contribute to the observed negative temperature–

precipitation correlation.” Differences in the relative contributions of these processes may explain some of the differences between the models.

L55: While I like the multivariate approach here, I suspect that the TX change is dominating over the Pr change in line with the findings of previous studies.

We have added a new section S8 that confirms the reviewer’s assertion: the Tx S/N is greater than the Pr S/N in almost all grid cells. We have added some clarifying text to the Results section “Factors contributing to departure intensification”:

“The strength of the projected climate change signal is larger in temperature than in precipitation in almost all grid cells (Supp. Info. S8). However, the precipitation trend determines the alignment of climate change with interannual variability.”

Figure 2b: I think the label should read “Historical + RCP4.5 30-yr running mean”

We originally excluded a reference to the “historical” run for brevity, but agree with this suggestion and have now included it in both Figure 1 and Figure 2.

L89: Is the Tx change always larger than the Pr change?

We have added a new section S8 “Relative departures of temperature and precipitation” to the supporting information to answer this question. We have provided an assessment of the difference in 2σ proportions of Tx and Pr in the 2021-2050 period (Figure S11). There are only a few locations where Pr departures are greater than Tx departures, and these occurrences are not consistent between models. Some of these occurrences persist into the 2071-2100 period.

Figure 3: I'm a bit confused as to exactly what is being shown here- is it the average departure between runs of the same model? Is the average a median or a mean (a mean-average is sensitive to outliers)? It would be useful to have an indication of the level of agreement between land area proportions between runs of the same model and different models (similar to King et al. 2015; Bador et al. 2016).

Important question. We have reworded the Figure 3 caption and added a supp info section S9 to clarify the calculation of the values being mapped. The calculation of maximum departure difference from the ensemble of runs for each model is illustrated in Figure 2b. First, we calculate the mean time series of univariate and bivariate 2-sigma proportion from the time series of each run; then we find the maximum departure difference between these mean time series. The alternate approach would be to take the mean of the maximum departure differences of each model run. We used the former approach and not the latter because 2-sigma proportion is quite noisy (due to calculation from a binary criterion); the maximum difference between noisier time series is greater, and hence the latter approach produces a biased (overestimated) maximum departure difference. A Figure 3 calculated from the latter approach is shown below: compared to Figure 3 in the manuscript, it clearly demonstrates the bias induced by taking the mean of the maximum departure differences of individual runs. The drawback of our approach is that it synthesizes all model runs into a single value, and therefore precludes the calculation of within-model variation in departure intensification (though see shaded regions of the 2-sigma proportion time series in Figures 2b-g for a sense of intramodel variation in a few grid cells of the CanESM2 model).

L124-127: This result contrasts with other studies which use an early baseline and already find indications of emergence (e.g. Mahlstein et al. 2012; King et al. 2015). A comment on this difference would be useful.

We agree that this opening statement to this paragraph was misleading because it was overly generic and have changed it to read:

“Bivariate climate departures are detected in tropical and subtropical regions of most climate models by the year 2020 (Figure 2 and Supp. Info. S3). This suggests that departure intensification may be detectable in the observational record at low latitudes, consistent with other studies that have detected emergence of a local warming signal (Mahlstein et al. 2012). Departure intensification is likely not yet detectable in the observational record of extratropical regions where the climate signal is yet to emerge from natural variability.”

As a side note, both Mahlstein et al. 2012 and King et al. 2015 used the Kolmogorov-Smirnov test to measure climate emergence, an approach which appears to be more sensitive than S/N. Mahlstein et al. 2012 provided a comparison to the S/N approach (their Figure 3b), and found that the emergences detected with the KS test were entirely undetected with $S/N > 1$. In addition to the S/N method being less sensitive, the 2-sigma proportion has lower sensitivity to small (< 0.5 sigma) climate shifts because increases in the frequency of 2-sigma anomalies on one tail of the distribution are partially balanced by decreases on the other tail (indicated by the sigmoidal shape of the response curves in Figure S7a). This is a useful, conservative, feature of the 2-sigma proportion but it does reduce the potential to detect departures and departure intensification in the early stages of emergence of the climate signal.

References:

Bador M, Terray L and Boé J 2016 Emergence of human influence on summer record-breaking temperatures

over Europe Geophys. Res. Lett. 43 404–12

Hawkins E and Sutton R 2012 Time of emergence of climate signals Geophys. Res. Lett. 39 L01702

King A D, Donat M G, Fischer E M, Hawkins E, Alexander L V, Karoly D J, Dittus A J, Lewis S C and Perkins S E 2015 The timing of anthropogenic emergence in simulated climate extremes Environ. Res. Lett. 10 094015

Mahlstein I, Hegerl G and Solomon S 2012a Emerging local warming signals in observational data Geophys. Res. Lett. 39 L21711

Reviewer #2 (Remarks to the Author):

Wetter summer can intensify departures from natural variability in a warming climate
by Colin R. Mahony and Alex J. Cannon

Reviewer: Milan Flach, Max Planck Institute for Biogeochemistry

*Suggestion for the Editor: *

The paper 'Wetter summer can intensify departures from natural variability in a warming climate' by Colin R. Mahony and Alex J. Cannon is well presented, well written, timely, and important.

The authors show that dependencies in earth system models between multiple variables (here: temperature and precipitation) cannot be neglected when computing the 'time of emergence' or departure from natural variability at each model location. Specifically they show, that many regions on Earth depart faster from natural variability in case one considers the multivariate correlations structure of temperature and precipitation. They provide sufficient evidence for their conclusions.

Technically, they use the Mahalanobis distance to the mean as a measure of multivariate departure from natural variability. The statistical analysis via Mahalanobis distance is sound as such. Although the technique dates back several decades ago and is common in other fields of science, it is not commonly used in this particular field of science to the best of my knowledge. Thus, the technique is novel for this particular application on CMIP5 models. The novel multivariate perspective furthermore offers advance to previous journal articles on this topic. In general, the manuscript allows to reproduce the results. There are only minor issues to improve understandability, which are described in detail below.

The paper is timely and interesting to others in the field as it goes in line with recent activity on multivariate extremes / compound events. I believe, that it sets new standards for future publications on the time of emergence / departure time, as it highlights the importance of the multivariate perspective on this particular topic.

I would recommend to accept the paper for publication in nature communications after the authors carefully addressed the following questions.

Best regards,
Milan Flach

*Major Comments to the Authors: *

1.) The definition of summer in the paper seems to be a bit too simplistic to me. Specifically, as already mentioned in the discussion (p.6, l.158), it hinders a straightforward interpretation of your results in the tropics (p.5, l.110). I was wondering, if it would be possible to change the definition of your summer (the warm period)

to be the hottest three month of tmax (comparable e.g. to (1)). This rather small change would make your statement much stronger in low latitude regions, which are very much affected by the intensification of departure.

We agree with the reviewer that a hottest-3 months analysis can be helpful, and we have provided a parallel set of results using this alternative definition of summer in supporting information section S10. We would prefer to maintain our current definition of summer in the main manuscript to maintain the connection to the many other studies that use a JJA/DJF definition of summer, and also due to the simplicity and transparency of the approach. The overall difference between the two approaches is subtle, though it is regionally important in the tropics as predicted by the reviewer. We believe that providing results for both definitions of summer will be informative for readers.

Thank you for pointing us towards Zscheischler & Seneviratne (2017)., a highly relevant related study which we have added to our references on compound events.

2.) You state in the introduction, that the environment is locally adapted to the climate in which it is living (p.1, paragraph 1/2). You are using runs of the CMIP5 models with historical natural variability to quantify the normal local variability within each gridcell. In case, one uses historical runs, one assumes that no further adaptation is taking place, which is not very plausible to my mind. Nevertheless, your approach is valid and commonly used. However, I was wondering about the time scale of local adaptation to climate. May be you could elaborate on this a bit further in the introduction. Defining a specific time scale for adaptation to the local climate would be another option for such kind of studies, which goes behind the scope of this paper, but nevertheless should be mentioned.

This is an excellent point, and applies more generally to the larger literature on time-of-emergence and climate departures. We have added a paragraph in the discussion to raise this issue. We agree that at any point in time, ecological and social systems will have adapted to some extent to historical climate changes, and that a pre-industrial baseline doesn't account for these adaptations. However, there is a counter-argument that many of these adaptations will have been disruptive and costly, and therefore should be accounted for as impacts of climate change. .

The added paragraph reads: "The impacts of climate departures are subject to the timescales over which maladaptation to unfamiliar local conditions is mitigated by gene flow, innovation, and other non-disruptive adaptive processes. In contrast to studies using a recent reference period(Giorgi and Bi 2009, Hawkins and Sutton 2012, Mora et al. 2013), our pre-industrial baseline precedes any local adaptation that has occurred during the industrial period, and may overstate the timing and magnitude of some of the disruptions associated with climate departures. The timescales of local adaptation are an important consideration in the assessment of the specific impacts of climate departures."

3.) On p.4, l 95, you are stating that large departure correlation goes in line with orthogonality to the natural mode of interannual variability. Would it be possible to provide further (quantitative) evidence for this finding as supplementary material? Probably, the second component of a principal component analysis (PCA) could serve as a measure of orthogonality to natural variability. PCA is technically very much related to / the same as the Mahalanobis distance to the mean.

We have made several modifications of the manuscript to quantify the orthogonality of the bivariate climate signal as a factor in departure intensification.

- Added Illustration of orthogonality to Figure 2b and quantified orthogonality in figures 2b-g;
- modified Figure 4 to include color-theming by orthogonality;
- added maps of the orthogonality of climate change to Supp Info Section S7;

- Restructuring of results section “Factors contributing to departure intensification” to emphasize correlation and orthogonality as the two key factors in departure intensification; and
- Added method for calculating orthogonality to methods section “Bivariate standardized anomalies.

We thank the reviewer for this suggestion, as the quantification of orthogonality is an important improvement to this paper.

4.) On p. 7, ll. 224-225 you are defining the univariate 2 sigma proportion as the maximum of the 2 sigma proportions of temperature and precipitation for computing the departure difference. May be it would improve understanding, if you add a (pseudo) formula here, like $\max(\text{prop}(Tx), \text{prop}(Pr))$, in case I understood it correctly. It would also be nice if you elaborate further on the reason for this specific choice. Furthermore other choices would be thinkable, like computing the 2 sigma proportion as sigma exceedances of either precipitation or temperature ($\text{prop}(Tx \text{ or } Pr)$), or computing the 2 sigma proportion of the marginal distributions of temperature or precipitation separately ($\text{prop}(Tx)$ or $\text{prop}(Pr)$). The latter choice would probably be very helpful to disentangle / attribute high departure differences to the univariate drivers. Thus, I would highly appreciate if you could provide some supplementary material on e.g. the latter alternative choice.

We have added two sections in the supporting information that address this comment: The first, “S9. Pseudocode for calculation of maximum departure difference,” describes and provides a rationale for the approach we took. This pseudocode complements the R code provided in the original package. The second section, “S8. Relative departures of temperature and precipitation,” demonstrates that temperature is almost exclusively driving the univariate departure, and therefore that the suggested alternative approach of calculating departure difference from the marginal distributions would likely have very subtle impact on the results.

Minor Comments to the Authors:

p.1, l. 3.: ‘interactions between climate variables’. The meaning of the word ‘interactions’ goes much further, than what you are actually looking at. I would favour a more neutral wording like ‘correlations’ or ‘dependencies’. (similar: p.4, l. 48; p. 5, l. 146; p.6, l. 173, 182, 184, 187)

We agree, and have changed the wording accordingly:

p.1, l. 3: changed “interactions” to “dependencies”

p.4, l. 48; We changed to “Interactions and dependencies between climate variables” because we are referring to the broader literature on compound events, where interactions may play a role.

p. 5, l. 146; rewrote the introductory sentence of the discussion as “We have demonstrated that a multivariate climate change trend can be stronger, relative to natural variability, than all of the individual trends of its component variables.”

p.6, l. 173; We kept this as “Interactions and dependencies among climate variables” because we are referring to the broader class of compound events.

p.6, 182; “~~We have demonstrated that variable interactions~~Decoupling of the climate change trend from the dominant historical mode of interannual variability can accelerate the rate at which locally unfamiliar climates develop”

p.6, 184; “Our study has focused on ~~interactions between~~ this effect in terrestrial summer temperatures and precipitation”

p.6, 187; “The potential for climate departures to be amplified ~~by variable interactions~~ in coupled climate variables is an important consideration...”

p.4, l. 87: ‘sigma dissimilarity, a novel method.’ The technique of ‘sigma dissimilarity’ translates anomaly scores of the multivariate Mahalanobis distance into percentiles of the theoretically received chi-squared distribution. Although I am in favour of the ‘sigma dissimilarity’ technique, calling this method ‘novel’ without a detailed literature review seems a bit exaggerated to me. The Mahalanobis distance is used since 1936 (2) and used for novelty / anomaly detection as Hotelling’s T² control chart since 1947 (3). The reviewer would be wondering, if nobody used percentiles of the chi-squared distribution in these decades. The reviewer would like to ask the authors to remove ‘novel’ from the sentence or to reword and clarify: what is particularly novel on the sigma dissimilarity technique?

We used the word “novel” in the context of calculating multivariate S/N and standardized anomalies. That said, we see that it can easily be misinterpreted as a more general claim of novelty. We concur with the reviewer that a more conservative wording is appropriate. We removed the word “novel” as suggested and included an explicit reference to Mahalanobis distance: “...sigma dissimilarity (Mahony et al. 2017), a parametric method of calculating multivariate S/N ratios and standardized anomalies using Mahalanobis distance (Mahalanobis 1936).”

p. 5, ll.132-134: This finding is particularly interesting for estimating departure intensification in (methodologically more simplistic) impact focussed studies on novel climates. However, whether higher correlations lead to a higher ‘risk’ in terms of impact on humans or biological systems is questionable / not a direct result of this study. It would require further research or might be better moved to the discussion part.

We changed the wording from “...at risk of...” to “...that are susceptible to...”

p. 6, l. 175: Did you actually try to detect the multivariate events by SPEI?

We did not. Moreover, this sentence is not correct, since PC2 is a synthetic univariate index that would indeed capture departure intensification quite well. We have changed the wording:

“The case of departure intensification illustrates that some compound events are ~~purely~~ primarily multivariate : ~~they cannot be detected~~ and can remain undetected in ~~by~~ univariate indices that synthesize temperature and precipitation, such as the standardized precipitation-evapotranspiration index and wet bulb temperature”.

p. 6, ll. 178-179 and 184: Misleading for my understanding is the ‘mechanism by which compound events to occur’, which you refer to. As far as I understood the study, it is focused on statistical correlations between precipitation and temperature and not on ‘mechanisms’ which would induce those kind of departures. Could you please elaborate further on what they specifically mean or rephrase the statement.

We changed the wording to “Our study demonstrates a form of compound extremes arising from historically unusual combinations of conditions. These compound anomalies can occur both as single-year events and long-term climatic shifts.”

p. 6, ll. 182-183: ‘We have demonstrated that variable interactions can accelerate the rate at which locally unfamiliar climates develop, which is a critical limitation on the ability of organisms and societies to adapt to climate change.’ The second part of this statement is not a direct finding of this study and requires a citation as such.

We softened the wording to “which may be a limitation on the ability of some organisms...” and added references to the literature that suggests this to be the case.

p. 6, l.190: Mentioning the reference period immediately rises the question: which reference period? It would probably be helpful for better understanding to include a reference to the later section on this topic.

We moved the description of the reference period variability to the beginning of the methods, above the description of standardized anomalies, to address this potential source of confusion.

p.7, l. 204: Please add a citation for the Mahalanobis distance ,e.g. (2) or (3) below, and/or reference number 31 from your manuscript (as more recent one).

We added a reference to Mahalanobis (1936) to our description of the sigma dissimilarity metric in the main text.

p. 7 / 8 / 9: References:

8. check page numbers

23. check page numbers / add DOI

31. discussion paper meanwhile published

33. check page numbers / add DOI

41. check page number and year

43. check page number

We have corrected these and a few more referencing errors

References:

(1) Zscheischler, J., & Seneviratne, S. I. (2017). Dependence of drivers affects risks associated with compound events. *Science Advances*, 3, e1700263)

(2) P. Mahalanobis, "On the generalised distance in statistics (vol.2, pp.49–55)," *Proceedings National Institute of Science, India*.

(3) Hotelling, H.: *Multivariate Quality Control - Illustrated by the Air Testing of Sample Bombsights*, in: *Techniques of Statistical Analysis*, edited by Eisenhart, C., Hastay, M. W., and Wallis, W. A., pp. 111–184, McGraw-Hill, New York, 1947.

Reviewer #3 (Remarks to the Author):

General Comments:

This is a very nice paper that clearly and thoroughly makes the case for considering pertinent variables in concert rather than in isolation when looking at changes in climate extremes. The authors focus on the relationship between summertime mean daily maximum temperature (Tx) and total precipitation (Pr), demonstrating that in many regions of the globe these variables are negatively correlated, with wet summers tending to be cool and dry summers tending to be hot. A warming climate, however, is pushing some regions towards warmer and wetter conditions. At these locations, the authors demonstrate that future conditions are likely to move beyond the historically-covered regions of the Tx-Pr parameter space more quickly than a climate change signal would emerge when considering either variable by itself. The authors made a good case for why these changes could have important consequences for many ecosystems. I believe this is a very strong paper and that *Nature Communications* is an excellent outlet for the material. I have listed a few minor suggestions for improvements below.

Specific Comments:

1. The introduction does a nice job of establishing the motivation for this study. You could draw further support by referring to examples of studies of future climate extremes that used variables depending on both

temperature and humidity, e.g. wet-bulb temperature (Pal and Eltahir, 2016; Im, Pal and Eltahir, 2017) or wet-bulb globe temperature (Knutson and Ploshay, 2016).

- Pal, J.S. and E.A.B. Elthair: Future temperature in southwest Asia projected to exceed a threshold for human adaptability. *Nature Climate Change*, 6, 197–200, doi:10.1038/nclimate2833, 2016.
- Im, Pal, and Eltahir: Deadly heat waves projected in the densely populated agricultural regions of South Asia. *Science Advances*, 02 Aug 2017; Vol. 3, no. 8, e1603322; DOI: 10.1126/sciadv.1603322.
- Knutson, T. R. and J. J. Ploshay: Detection of anthropogenic influence on a summertime heat stress index. *Climatic Change*, 138, doi:10.1007/s10584-016-1708-z , 2016.

We thank the reviewer for these suggestions and have added a reference to wet-bulb temperature (Knutson and Ploshay 2016) in the discussion.

2. Figure 1: “historicalNat” hasn’t been defined yet. I think it is only defined in the methods. You should mention that Figure S7 shows results from the individual models.

We opted to simplify “historicalNat runs” to “internal variability” for the purpose of maintaining brevity in the caption. We added “(see Figure S9 for single-model correlations)” to the caption.

3. Figure 2: The specific examples of different possibilities (low Tx-Pr correlation, etc) are great, and the text describing Figure 2 very clearly explains the three relevant factors for departure intensification. The caption should mention the time period captured by the map in plot a. What does negative number of years for the red lines mean? According to the y-axis label they are a number of years (out of 30) divided by 30. If this is not correct, please clarify. In the methods you describe how the departure difference can be negative, but not how the 2σ proportion can be negative.

We clarified “...measured as maximum departure difference during the 1880-2100 period.”

If we understand the reviewer’s concern correctly, the dashed line can be confused as zero on the y-axis. The reviewer is correct that 2σ proportion cannot be negative. We have added the following sentence to the caption to clarify: “The dashed line in plots b-g indicates the expected 2σ proportion of 0.046 in a stationary climate with Gaussian (normal) variability.”

4. Line 164: Your results only show variability correlations stronger than -0.5, not +/-0.5. I understand that you are leading into talking about other variable pairs, but perhaps you can rephrase this sentence so it is cleaner.

We have removed the reference to positive correlations and will rely on astute readers to infer that the effect can be expected to apply to positive as well as negative correlations, subject to orthogonality of the climate change trajectory.

5. Line 219: I think you should more explicitly indicate how you compute that a 3σ exceedance is a 1-in-370-year event.

We have clarified as “(an $\sim 1/(1-0.9973) = 1\text{-in-370-year exceedance}$)”

The supplementary materials provide details that may be of interest to certain readers and enhance the rigor and thoroughness of the study. I believe they are correctly placed in the supplement.

Reviewers' Comments:

Reviewer #1:

None

Reviewer #2:

Remarks to the Author:

I believe that the article has the potential to become an important reference in the field standing out through its consequent and novel multivariate perspective on the topic coming in line with very relevant results also for a broader community (departure intensification due to the multivariate perspective). The authors successfully addressed all my previous comments. In addition, they produced a second line of evidence for their findings based on another definition of summertime. The paper can be published after changing the following (very minor) two issues.

Milan Flach

1.) Check reference (substitute page number):

34. Trenberth, K. E., and D. J. Shea, Relationships between precipitation and surface temperature, *Geophys. Res. Lett.*, 32, L14703 (2005)

2.) Please carefully check the understandability / meaning of the following sentence from Supplementary S9. It might be very difficult to understand without prior knowledge on the review process.

"The univariate 2σ proportion is calculated from 2σ proportions of Tx and Pr because the alternate approach of selecting the maximum anomaly of either Tx or Pr in each year would result in an overestimate of the univariate 2σ proportion."

Response to reviews

We would like to thank the reviewer for his additional review. We have described the changes we have made and any responses to comments below in blue font.

Reviewers' comments:

Reviewer #2 (Remarks to the Author):

I believe that the article has the potential to become an important reference in the field standing out through its consequent and novel multivariate perspective on the topic coming in line with very relevant results also for a broader community (departure intensification due to the multivariate perspective). The authors successfully addressed all my previous comments. In addition, they produced a second line of evidence for their findings based on another definition of summertime. The paper can be published after changing the following (very minor) two issues.

Milan Flach

1.) Check reference (substitute page number):

34. Trenberth, K. E., and D. J. Shea, Relationships between precipitation and surface temperature, *Geophys. Res. Lett.*, 32, L14703 (2005)^[1]_{SEP}

The citation for this article is correct, though we acknowledge that a more complete reference would provide the DOI in addition to the article number. We will add the DOI to all online-only articles if requested by the editor.

2.) Please carefully check the understandability / meaning of the following sentence from Supplementary S9. It might be very difficult to understand without prior knowledge on the review process.

"The univariate 2σ proportion is calculated from 2σ proportions of Tx and Pr because the alternate approach of selecting the maximum anomaly of either Tx or Pr in each year would result in an overestimate of the univariate 2σ proportion."

In addition to being difficult to understand, this sentence is not necessary. We have deleted it.